# Hormesis Responses of Photosystem II in *Arabidopsis thaliana* under Water Deficit Stress

**DOI:** 10.3390/ijms24119573

**Published:** 2023-05-31

**Authors:** Ilektra Sperdouli, Georgia Ouzounidou, Michael Moustakas

**Affiliations:** 1Department of Botany, Aristotle University of Thessaloniki, GR-54124 Thessaloniki, Greece; ilektras@bio.auth.gr; 2Institute of Plant Breeding and Genetic Resources, Hellenic Agricultural Organization-Dimitra, GR-57001 Thessaloniki, Greece; 3Institute of Food Technology, Hellenic Agricultural Organization-Dimitra, GR-14123 Lycovrissi, Greece; geouz@yahoo.gr

**Keywords:** oxidative stress, reactive oxygen species (ROS), excess excitation energy, photoinhibition, electron transport, redox status, anthocyanins, photoprotection, acclimation, leaf developmental stage

## Abstract

Since drought stress is one of the key risks for the future of agriculture, exploring the molecular mechanisms of photosynthetic responses to water deficit stress is, therefore, fundamental. By using chlorophyll fluorescence imaging analysis, we evaluated the responses of photosystem II (PSII) photochemistry in young and mature leaves of *Arabidopsis thaliana* Col-0 (cv Columbia-0) at the onset of water deficit stress (OnWDS) and under mild water deficit stress (MiWDS) and moderate water deficit stress (MoWDS). Moreover, we tried to illuminate the underlying mechanisms in the differential response of PSII in young and mature leaves to water deficit stress in the model plant *A. thaliana*. Water deficit stress induced a hormetic dose response of PSII function in both leaf types. A U-shaped biphasic response curve of the effective quantum yield of PSII photochemistry (Φ*_PSII_*) in *A. thaliana* young and mature leaves was observed, with an inhibition at MiWDS that was followed by an increase in Φ*_PSII_* at MoWDS. Young leaves exhibited lower oxidative stress, evaluated by malondialdehyde (MDA), and higher levels of anthocyanin content compared to mature leaves under both MiWDS (+16%) and MoWDS (+20%). The higher Φ*_PSII_* of young leaves resulted in a decreased quantum yield of non-regulated energy loss in PSII (Φ*_NO_*), under both MiWDS (−13%) and MoWDS (−19%), compared to mature leaves. Since Φ*_NO_* represents singlet-excited oxygen (^1^O_2_) generation, this decrease resulted in lower excess excitation energy at PSII, in young leaves under both MiWDS (−10%) and MoWDS (−23%), compared to mature leaves. The hormetic response of PSII function in both young and mature leaves is suggested to be triggered, under MiWDS, by the intensified reactive oxygen species (ROS) generation, which is considered to be beneficial for activating stress defense responses. This stress defense response that was induced at MiWDS triggered an acclimation response in *A. thaliana* young leaves and provided tolerance to PSII when water deficit stress became more severe (MoWDS). We concluded that the hormesis responses of PSII in *A. thaliana* under water deficit stress are regulated by the leaf developmental stage that modulates anthocyanin accumulation in a stress-dependent dose.

## 1. Introduction

Water deficit is among the major limiting factors for plant growth, affecting several physiological and biochemical processes of plants [1,2,3,4]. As a consequence of climate change, water deficit stress episodes are expected to increase in frequency, intensity, and duration, reducing plant productivity [5,6,7,8]. Water scarcity impairs plant cell division, elongation, and differentiation; decreases plant growth; and impairs osmotic adjustment, causing loss of turgor [9,10,11,12]. Water deficit results in reduced water uptake by plants and, consequently, restricted nutrients for plant growth and development [13,14]. Moreover, water deficit stress conditions significantly constrain photosynthesis by stomatal closure to reduce excessive water loss via transpiration, thus limiting CO_2_ assimilation and increasing leaf membrane lipid peroxidation [2,8,15,16,17]. In the light reactions of photosynthesis, which take place in the thylakoid membranes of the photosynthetic apparatus, NADPH and ATP are generated, which are mandatory in the Calvin–Benson–Bassham cycle for carbon dioxide fixation to produce carbohydrates [18,19,20]. Thus, reduction in carbon fixation results in excess light energy that cannot be fully utilized by plants [21,22,23]. This surplus light energy has the potential for damage if it is transferred to oxygen, producing photo-oxidative injury and reducing the photosynthetic efficiency, a phenomenon recognized as photoinhibition [21,24,25,26,27]. The light energy that cannot be used in photochemistry can over-reduce the electron transport chain in the thylakoid membrane and lead to the creation of reactive oxygen species (ROS) [21,28].

Plant leaves dissipate the excess absorbed light energy as heat by the non-photochemical quenching (NPQ) mechanism to protect the photosynthetic apparatus, and especially photosystem II (PSII), from the photo-oxidative damage [21,29]. During water deficit stress, the excess light energy is enhanced, and this excess excited light energy must be dissipated as heat by the NPQ mechanism to avoid ROS and oxidative damage [4,23,28,30,31]. However, ROS that are created in plants under environmental stress conditions are not only harmful but can also be beneficial since they are involved in stress defense responses [32,33,34]. Thus, ROS produced in chloroplasts, besides generating oxidative stress, also present a vital biological function as redox signaling molecules that translate information from the environment [35,36,37,38,39,40].

Anthocyanins that increase in response to various environmental stresses such as drought, high light, nutrition, heavy metals, temperature, and wounding, can act as ROS scavengers to protect plants from oxidative stress, inhibiting in general lipid peroxidation [38,39,41,42,43,44]. Anthocyanins prevent or inhibit molecular oxidation depending on their –OH substituents and their ring orientation [45]. Sugar is the precursor of anthocyanin [46], and exogenous proline has been reported to induce soluble sugar accumulation [47]. An interaction of proline, sugars, and anthocyanins was found to alleviate drought stress effects in *A. thaliana* PSII function and contribute to an acclimation process [48]. An increased anthocyanin accumulation under high ROS circumstances, in return, assists in ROS scavenging [38]. In Arabidopsis, the role of anthocyanins in photoprotection has been described both via light attenuation and as antioxidants [49,50]. However, the role of anthocyanins in photosynthesis and photoprotective mechanisms remains controversial [39,51].

Examining the negative impact of water deficit stress on growth and yield will require estimating the mode under which it influences photosynthesis; thus, examining leaf photosynthesis remains crucial [52]. Photosynthetic efficiency in *A. thaliana* was higher under moderate water deficit stress (MoWDS) than under mild water deficit stress (MiWDS) [53]. Yet, plants that were described to be tolerant to severe water deficit stress (SWDS) did not present tolerance under MiWDS [54]. Exploring the molecular and physiological strategies that plants apply to grow with limited water is, therefore, fundamental [55].

Plants need to maintain an equilibrium between the capture of light energy, its allocation to the reaction centers, the production of ATP and NADPH, and their consumption for carbohydrates synthesis [4,22,56,57]. The reaction of plants to water deficit stress is the closure of their stomata to eliminate water loss, which results in a reduced entrance of CO_2_ and a reduced synthesis of carbohydrates, and thus, in a lesser demand for ATP and NADPH [4,58,59].

Plant responses to a disruption of homeostasis caused by a low stress level display an overcompensation reaction that results in a hormetic stimulation [60,61]. Hormesis is the positive effect of a low dose of a stressor on living organisms that is followed by a negative (opposing) effect at a higher dose of the same stressor [62]. Still, a hormetic response appears also with a low dose or short time inhibition and a higher dose or longer time stimulation [63]. Hormesis describes a biphasic dose–response relationship [64,65,66]. For example, the hormetic responses of PSII photochemistry to Cd were described by an inverted U-shaped response curve with a low-dose or short-time exposure stimulation and a high-dose or longer-duration exposure inhibition [67]. A hormetic stimulation of PSII function under biotic stress was proposed to be activated by ROS [68]. Recently, the phenomenon of hormesis was shown to occur in a number of organisms independent of the kind of stressor or the physiological process examined [27,61,69,70,71,72,73,74].

Despite various studies implicating anthocyanins in environmental stress responses, it is basically unknown how the regulation of water deficit stress tolerance is related to the leaf development stage and to anthocyanin accumulation [23,75]. However, the leaf developmental stage is critical for the photosynthetic response to water deficit stress [4,23]. Thus, in order to address this issue, we chose as experimental material the model plant *A. thaliana* to explore how anthocyanin accumulation and lipid peroxidation are modulated in young and mature leaves, at the onset of water deficit stress (OnWDS) and under mild water deficit stress (MiWDS) and moderate water deficit stress (MoWDS). In addition, we set out to explore if anthocyanin accumulation, in response to water deficit stress, influences oxidative stress and if this is affecting PSII photochemistry in both leaf types.

## 2. Results

### 2.1. Leaf Water Content in Young and Mature Leaves under Water Deficit Stress

The leaf water content (%) in both young and mature leaves decreased significantly by the onset of water deficit stress (OnWDS), being decreased further under mild water deficit stress (MiWDS) and showing the lowest values under moderate water deficit stress (MoWDS) (Table 1). However, under both MiWDS and MoWDS, young leaves retained significantly higher water content compared to mature leaves (Table 1).

### 2.2. Oxidative Stress in Young and Mature Leaves under Water Deficit Stress

The level of lipid peroxidation, measured as malondialdehyde (MDA) content, and representing the oxidative stress, increased significantly only in mature leaves with the OnWDS. At MiWDS, both young and mature leaves showed higher oxidative stress since MDA content declined significantly in both leaf types at MoWDS, compared to MiWDS (Table 1).

Under control conditions, mature leaves had lower oxidative stress compared to young, while young leaves had lower oxidative stress, compared to mature, under both MiWDS (−17%) and MoWDS (−7%) (Table 1).

### 2.3. Anthocyanin Accumulation in Young and Mature Leaves under Water Deficit Stress

Anthocyanin accumulation in both leaf types of *A. thaliana* increased in a dose–response mode to water deficit stress (Table 1). Under control conditions and the OnWDS, mature leaves had higher anthocyanin content compared to young leaves, while young leaves accumulated more anthocyanin compared to mature, under both MiWDS (+16%) and MoWDS (+20%) (Table 1). Anthocyanin content at MoWDS increased 33-fold in young leaves and 22-fold in mature, compared to controls (well-watered).

### 2.4. Maximum Efficiency of Photosystem II Photochemistry in Young and Mature Leaves under Water Deficit Stress

The maximum efficiency of PSII photochemistry (F*v*/F*m*) decreased significantly by the OnWDS in both young and mature leaves, showing the highest decrease under MiWDS, while it increased in both leaf types under MoWDS, compared to MiWDS (Table 1). Under optimum conditions (control), mature leaves had higher PSII maximum efficiency (F*v*/F*m*) compared to young, while young leaves had higher F*v*/F*m* values under both MiWDS (+8%) and MoWDS (+4%) (Table 1).

### 2.5. Light Energy Use in Photosystem II of Young and Mature Leaves under Water Deficit Stress

The absorbed light energy by the antenna is partitioned to PSII photochemistry (Φ*_PSII_*), regulated non-photochemical energy loss in PSII (Φ*_NPQ_*), and non-regulated energy loss in PSII (Φ*_NO_*), with the sum of all to be equal to one [76]. The effective quantum yield of PSII photochemistry (Φ*_PSII_*) decreased significantly by the OnWDS in mature leaves (Figure 1) but did not change in young leaves (Figure 2). Φ*_PSII_* showed the highest decrease in both young and mature leaves under MiWDS, while there was an increase in both leaf types under MoWDS compared to MiWDS (Figure 1 and Figure 2).

The quantum yield of regulated non-photochemical energy loss in PSII (Φ*_NPQ_*) decreased under water deficit stress in mature leaves showing the lowest values under MiWDS (Figure 1). Owing to the decreases in Φ*_PSII_* and Φ*_NPQ_* in mature leaves under water deficit stress, the quantum yield of non-regulated energy loss in PSII (Φ*_NO_*) increased in mature leaves under water deficit stress, showing the highest values under MiWDS (Figure 1). In young leaves, due to the photoprotective increase in Φ*_NPQ_* under water deficit stress, compared to control, Φ*_NO_* decreased by the OnWDS but increased under MiWDS and MoWDS (Figure 2). Nonetheless, Φ*_NO_* in young leaves remained significantly lower than mature leaves under both MiWDS (−13%) and MoWDS (−19%) (Figure 1 and Figure 2).

### 2.6. Open Reaction Centers of Photosystem II and their Efficiency in Young and Mature Leaves under Water Deficit Stress

The fraction of open PSII reaction centers (q*p*), that is, the redox state of PSII, decreased significantly by the OnWDS in mature leaves but did not change in young leaves (Table 2), showing the highest decrease in both young and mature leaves under MiWDS, while it increased in both leaf types under MoWDS, compared to MiWDS (Table 2). Young leaves retained a more oxidized PSII redox state than mature leaves, by 27% at the OnWDS, by 40% at MiWDS, and by 36% at MoWDS (Table 2). The efficiency of the open PSII reaction centers (F*v*’/F*m*’) did not change with the OnWDS, in both leaf types, compared to controls, while under both MiWDS and MoWDS decreased in young leaves, but in mature leaves decreased only during MiWDS and remain similar to controls during MoWDS (Table 2). Under the same water deficit stress treatment, F*v*’/F*m*’ did not differ between young and mature leaves (Table 2).

### 2.7. Electron Transport Rate and Excess Excitation Energy in Photosystem II in Young and Mature Leaves under Water Deficit Stress

The electron transport rate (ETR) decreased significantly by the OnWDS in mature leaves (Table 2) but did not change in young leaves (Table 2), showing under MiWDS the greatest reduction in both leaf types. During MoWDS, ETR increased in both leaf types compared to MiWDS (Table 2). Young leaves showed higher ETR compared to mature, by 30% at the OnWDS, 46% at MiWDS, and 40% at MoWDS (Table 2).

The excess excitation energy (EXCEE) increased significantly in mature leaves with the OnWDS but did not change in young leaves compared to controls (Table 2), showing the highest increase, compared to controls, in both young and mature leaves under MiWDS (Table 2). Under MoWDS, EXCEE decreased in both leaf types compared to MiWDS (Table 2). Nevertheless, EXCEE in young leaves remained significantly lower than mature leaves under both MiWDS (−10%) and MoWDS (−23%) (Table 2).

### 2.8. Correlation Analysis between Light Energy Use and Excitation Pressure in Young and Mature Leaves under Water Deficit Stress

A linear regression analysis was performed among some chlorophyll fluorescence parameters at a light intensity of 136 μmol photons m^−2^ s^−1^, and the level of oxidative stress was evaluated by malondialdehyde (MDA). The quantum yield of non-regulated energy loss in PSII (Φ*_NO_*) was significantly positively correlated to the level of excitation pressure (1-q*_p_*) in young and mature *A. thaliana* leaves (Figure 3a). The effective quantum yield of PSII photochemistry (Φ*_PSII_*) in both leaf types, under control conditions and all water deficit stress treatments, was significantly negatively correlated to the level of excitation pressure (1-q*_p_*) (Figure 3b).

Under control conditions and all water deficit stress treatments, in both leaf types, the level of excitation pressure (1-q*_p_*) was significantly positively correlated to the level of oxidative stress, measured as malondialdehyde (MDA) (Figure 3c), and to the level of excess excitation energy (EXCEE) (Figure 3d).

### 2.9. Correlation Analysis of Light Energy Use and Oxidative Stress in Young and Mature Leaves under Water Deficit Stress

The level of excess excitation energy (EXCEE), in both leaf types, under control conditions and all water deficit stress treatments, was significantly positively correlated to the quantum yield of non-regulated energy loss in PSII (Φ*_NO_*) (Figure 4a) and the level of oxidative stress (MDA) (Figure 4c), but it was significantly negatively correlated to the effective quantum yield of PSII photochemistry (Φ*_PSII_*) (Figure 4b). On the other hand, the level of oxidative stress (MDA) was significantly negatively correlated to the maximum efficiency of PSII photochemistry (F*v*/F*m*) (Figure 4d).

### 2.10. Hormetic Responses of Photosystem II in Young and Mature Leaves under Water Deficit Stress

The response of PSII function to water deficit stress in both young and mature leaves can be depicted by a U-shaped biphasic curve. Φ*_PSII_* showed the highest decrease in both young and mature leaves under MiWDS but increased in both leaf types under MoWDS, showing the U-shaped hormetic dose response (Figure 5).

## 3. Discussion

Among all destructive processes that are adversely affecting arable lands and food production, drought is the most significant risk to agriculture [4,77]. Water deficit stress significantly decreases photosynthetic activity as a result of stomatal closure that reduces CO_2_ availability, with a consequence of a decreased electron transport rate (ETR) [2,12,15,16]. In such cases, the absorbed light energy exceeds what can be used for photochemistry, resulting in excessive ROS accumulation that can harm the chloroplast and particularly damage photosystem II (PSII) [8,12,34,78,79,80].

The ETR, and also Φ*_PSII_*, in both young and mature leaves, decreased under both mild water deficit stress (MiWDS) and moderate water deficit stress (MoWDS) compared to controls (Figure 1 and Figure 2). Yet, in both leaf types, Φ*_PSII_*, and as a consequence, ETR, were significantly lower under MiWDS compared to MoWDS (Figure 1 and Figure 2). These results verify previous data that have shown that Φ*_PSII_* was higher under MoWDS compared to MiWDS in *A. thaliana* [53] and that photosynthetic function was better under MoWDS than under MiWDS in young leaves compared to mature [23,75]. In accordance with these results, young leaves in our experiment had higher ETR and Φ*_PSII_* compared to mature leaves under both MiWDS and MoWDS (Table 2, Figure 1 and Figure 2). A decrease in Φ*_PSII_* can be attributed either to a decline in the fraction of open PSII reaction centers (q*p*) or/and to a decreased efficiency of these centers (F*v*’/F*m*’) [81]. The significantly higher Φ*_PSII_* in young leaves compared to mature, under water deficit stress, was possibly due to a significantly higher fraction of open PSII reaction centers (q*p*) (Table 2) since the efficiency of the open reaction centers (F*v*’/F*m*’) did not differ between them under the same water deficit treatment (Table 2).

Under control conditions, mature leaves had lower oxidative stress and higher PSII maximum efficiency (F*v*/F*m*) compared to young leaves, while under both MiWDS and MoWDS, young leaves had lower oxidative stress and higher PSII maximum efficiency (F*v*/F*m*), compared to mature ones (Table 1), thus suggesting a lower degree of photoinhibition [26,27,82,83,84,85] for young leaves. The higher resistance of young leaves to photoinhibition (Table 1) under water deficit stress, compared to mature ones, was possibly due to a higher oxidized state of the PQ pool (q*p*) (Table 2). The reduction status of the plastoquinone pool has been shown in many cases to be the most sensitive and suitable indicator to probe photosynthetic function, determine the impact of environmental stresses on plants, and select drought-tolerant cultivars [8,12,36,53,57]. A significant positive correlation between excitation pressure (1-q*_p_*) and excess excitation energy (EXCEE) to oxidative stress was noticed (Figure 4). The decreased excess excitation energy (EXCEE) in young leaves compared to mature (Table 2) during water deficit treatments indicates an improved PSII efficiency.

Water deficit stress may not affect the photosynthetic function in a uniform way [53], and thus photosynthetic performance may be extremely heterogeneous at the leaf surface [4,63], as it was observed in both young and mature *A. thaliana* leaves under MiWDS (Figure 1 and Figure 2). The decreased Φ*_PSII_* in both leaf types, under both MiWDS and MoWDS, triggered an excess excitation energy (Table 2) that over-reduced the redox state of the PQ pool (Table 2), closing a fraction of open PSII reaction centers (q*p*) (Table 2) [38,86]. Nevertheless, young leaves had a significantly higher fraction of open PSII reaction centers (Table 2) and lower oxidative stress (Table 1) compared to mature leaves. This lower oxidative stress in young leaves compared to mature ones was associated with a higher anthocyanin accumulation (Table 1).

Under all water stress treatments, the regulated non-photochemical energy loss as heat (Φ*_NPQ_*) in mature leaves was strongly limited compared to controls (Figure 1), possibly due to damage to PSII structure and functionality. Insufficient dissipation of excess excitation energy can result in photo-oxidative stress, which is initiated by excess electrons in the photosynthetic light reactions, leading to subsequent ROS creation [53,87,88,89]. This seems to be the principal reason for the higher level of lipid peroxidation in mature *A. thaliana* leaves compared to young ones, in which thermal dissipation (Φ*_NPQ_*) was not decreased compared to the control (Figure 2).

Anthocyanin accumulation was higher in young leaves compared to mature leaves (Table 1) and was possibly associated with reduced oxidative damage (Table 1). The ability of anthocyanins to act as ROS scavengers, and thus to limit the excitation pressure and the excess excitation energy, and also to lower the susceptibility to photoinhibition has previously been mentioned [23,38,48]. Accordingly, a higher anthocyanin concentration in red apple peels resulted in a lower ROS accumulation [90]. Anthocyanins may protect plants from photoinhibition [41] and act directly as antioxidants [38].

The differential water deficit response to drought stress of barley leaves was shown to underly the activation of an appropriate defense response [91]. In our study, the differential water deficit response of young and mature leaves was triggered by a distinct defense response that resulted in differential oxidative stress in young and mature leaves. This was possibly due to a differential ROS accumulation in mature and young leaves that differentially modified the redox state of the plastoquinone pool (Table 2).

The hormetic response of PSII photochemistry under water deficit stress is described by a U-shaped response curve with an inhibition of PSII photochemistry at low-dose water deficit stress (MiWDS) and an increase in PSII photochemistry at higher-dose stress (MoWDS) [55,61]. The form of this hormetic U-shaped dose–response relationship is determined by various factors, such as the response endpoint measured, the time of measurement, and the spacing of the doses included in the experiment [62,64,70,73].

The lower competence of mature leaves compared to young ones to utilize the absorbed light energy for photochemistry (Φ*_PSII_*), or to safely dissipate it as heat (Φ*_NPQ_*) under MiWDS and MoWDS, developed an increased Φ*_NO_* and, consequently, an increased triplet chlorophyll state (^3^Chl*) population that created the reactive ^1^O_2_ [92,93,94]. Therefore, the probability of ^1^O_2_ formation can be calculated by Φ*_NO_* [95,96]. A decreased Φ*_NO_* in young leaves (Figure 2) compared to mature leaves (Figure 1) suggests a better photoprotection and reveals a lower ^1^O_2_ production [94,95,97].

The differentially modulated ROS accumulation in young and mature leaves, as judged from the ^1^O_2_ production, activated possibly differential stress response transduction pathways. ROS signaling is reported to stimulate anthocyanin accumulation, and as a response, anthocyanin accumulation modulates ROS production and supports photosynthetic efficiency [98]. Stimulation of anthocyanin biosynthesis has been suggested to result from chloroplast signaling transduction via the redox state of the electron transport chain [38,99,100]. It is suggested that the differentially modulated ROS accumulation in young and mature leaves differentially modified the redox state of the plastoquinone pool under MiWDS and generated a ROS operational signal to stimulate anthocyanin biosynthesis (Table 1). A “signal modulation” theory for anthocyanin function under environmental stress has long ago been proposed [101].

Plants must sense when the chloroplasts are stressed and induce operational signals to accomplish it [102]. ROS generated in chloroplasts are not only producing oxidative stress but also confer important biological function as redox signaling [38,103,104,105]. The signal transduction routes that are induced by the redox state of the PQ pool also involve a plant acclimation mechanism [106]. ROS also play an essential role in the regulation of leaf development and act as signaling molecules that translate information from the environment [107,108,109]. Redox homeostasis and signaling play a role in response to abiotic stressors such as drought, low temperature, and osmotic stress [107,110,111]. Leaf veins mediate systemic ROS signaling [112] during both biotic and abiotic stress conditions to initiate defense stress responses [34,112,113].

The simultaneous increase, in both leaf types under MiWDS, of Φ*_NO_* (Figure 1 and Figure 2), a measure of the singlet-excited oxygen (^1^O_2_) generation [95,96], and of the lipid peroxidation, evaluated by MDA (Table 1), suggests an increase in ROS production which was accompanied by a reduced redox status of the plastoquinone pool (q*p*) (Table 2). The redox state of the plastoquinone pool (q*p*) also comprises a mechanism of plant acclimation to environmental stresses by regulating photosynthetic gene expression [114,115,116,117] and is of exceptional significance for antioxidant defense and signaling [118]. For instance, the reduced redox status of the plastoquinone pool was proposed to mediate stomatal closure, offering acclimation to Cd exposure [63,119]. Recently, it has been frequently proposed that the reduction status of the plastoquinone pool (q*p*) is the most sensitive and suitable indicator to assess PSII functionality under many biotic and abiotic stress factors [4,12,57,120,121,122]. Under any biotic or abiotic stress factor, the photoprotective dissipation of excess light energy as heat (NPQ) can be considered efficient only if the fraction of open reaction centers (q*p*) remains the same as that in control conditions [57,123,124]. In our experiment, the photoprotective NPQ mechanism was sufficient only in young leaves at the OnWDS (Table 2). However, the photoprotective heat dissipation was more efficient under water stress treatments in young leaves compared to mature ones (Table 2).

Though ROS were originally assumed to be toxic by-products that must be scavenged to avoid oxidative damage, it is now generally accepted that ROS are functioning as both toxic by-products as well as important signal transduction molecules that are involved in stress sensing and signaling [33,113,125]. Our results confirm the statement that ROS at basal levels are essential to sustain life, while an intensified ROS creation is considered to be beneficial for activating the molecular mechanisms of plant stress tolerance [33,34,126,127]. Current advances connect ROS signaling with other essential stress-response signal transduction pathways to establish defense mechanisms and plant resilience to biotic and abiotic stresses [127,128,129]. We propose that the redox status of the plastoquinone pool (q*p*) under MiWDS (Table 2), mediated by ROS, triggered the acclimation response of young leaves to MoWDS. Yet, it can be concluded that the intensified ROS production at MiWDS in both leaf types, as judged from the increased lipid peroxidation evaluated by MDA (Table 1) and the increased ^1^O_2_ generation estimated by the increased Φ*_NO_* (Figure 1 and Figure 2), can be considered as beneficial for activating defense stress responses [127,128,129].

## 4. Materials and Methods

### 4.1. Plant Material and Growth Conditions

*Arabidopsis thaliana,* which has been documented as the model organism for investigation in plant biology [130], was used as plant material. *A. thaliana* (L.) Heynh. (Col-0) seeds obtained from Nottingham Arabidopsis Stock Centre (NASC) (Nottingham, UK) were sown on a soil and peat mixture in a growth chamber [114]. For uniform germination, the seeds were incubated for 2 days at 4 °C before sowing on soil. The germinated seedlings of *A. thaliana* ecotype Columbia (Col-0) were grown in a growth chamber with a long day photoperiod 14 h/10 h, temperature 23 ± 1/20 ± 1 °C day/night, humidity 45 ± 5/60 ± 5% day/night, and light intensity of 130 ± 10 μmol photons m^−2^ s^−1^. Two developmental leaf stages were used for the measurements: mature leaves with a length of about 4.1 ± 0.5 cm and young leaves from the middle of the leaf rosette with 1.5–2 cm length [4,23,75].

### 4.2. Water Deficit Stress and Soil Water Status

Water deficit stress was imposed by withholding irrigation on 4-week-old Arabidopsis plants as described earlier [23]. Four different watering regimes were examined: control (i.e., optimal water availability), onset of water deficit stress (OnWDS, by withholding irrigation till maintaining 95–96% soil volumetric water content (SWC) of control plants), mild water deficit stress (MiWDS, by withholding irrigation till maintaining 66–68% SWC of control plants), and moderate water deficit stress (MoWDS, by withholding irrigation till maintaining 50–52% SWC of control plants).

Soil volumetric water content (SWC), measured as described previously [131], was estimated in m^3^ m^−3^ by the soil moisture sensor 5TE (Decagon Devices, Pullman, WA, USA) equipped with the read-out device ProCheck (Decagon Devices, Pullman, WA, USA).

### 4.3. Water Content of Young and Mature A. thaliana Leaves

The water content of young and mature leaves was determined by the electronic moisture balance (MOC120H, Shimadzu, Tokyo, Japan) as described previously [53]. The two developmental leaf stages that were examined were developing young leaves and fully developed mature ones.

### 4.4. Oxidative Stress Evaluation

Oxidative stress was estimated by measuring lipid peroxidation, which was evaluated by malondialdehyde (MDA) content with 2-thiobarbituric acid (TBA) [132] reaction as follows:[(Abs 532_+TBA_) − (Abs 600_+TBA_) − (Abs 532_−TBA_−Abs 600_−TBA_)] = A
[(Abs 440_+TBA_ − Abs 600_+TBA_) 0.0571] = B
where 532 nm is the maximum absorbance of the TBA–MDA complexes, 600 nm is the correction factor for non-specific turbidity, and 440 nm is the correction factor for sucrose interference.

MDA equivalents were estimated in nmol mL^−1^ from the following equation:(A − B)/157,000 × 10^6^, where 157,000 is the molar extinction coefficient for MDA.

### 4.5. Anthocyanin Determination

Anthocyanins were extracted from leaf discs with a methanol extraction buffer (containing 1% HCl) as described previously [48,133]. After centrifugation, absorption spectra were calculated at 530 and 657 nm. Relative anthocyanin content was quantified as absorbance cm^−2^ by the equation A_530_ – 0.25 × A_657_, using a PharmaSpec UV-1700 spectrophotometer (Shimadzu, Tokyo, Japan).

### 4.6. Chlorophyll Fluorescence Imaging Analysis

Chlorophyll fluorescence measurements were performed in vivo by the pulse-amplitude modulation (PAM) method using an IMAGING-PAM fluorometer (Heinz Walz GmbH, Effeltrich, Germany), as described previously [39]. The chlorophyll fluorescence parameters were estimated by Imaging Win V2.41a software (Heinz Walz GmbH, Effeltrich, Germany) and are described in Appendix A. The actinic light (AL) that was used was 136 μmol photons m^−2^ s^−1^, similar to the growth light of plants. Representative color-coded images, at 136 μmol photons m^−2^ s^−1^ AL, of the effective quantum yield of PSII photochemistry (Φ*_PSII_*), the quantum yield of regulated non-photochemical energy loss in PSII (Φ*_NPQ_*), and the quantum yield of non-regulated energy loss in PSII (Φ*_NO_*), are shown in order to reveal in both leaf developmental stages the whole leaf response at all treatments.

### 4.7. Statistical Analysis

Pairwise differences were analyzed with independent samples *t*-test using the IBM SPSS Statistics for Windows version 28 at a *p* < 0.05 level [113]. A linear regression analysis was also performed [48].

## 5. Conclusions

In this study, we revealed that PSII responses to water deficit stress, in both young and mature *A. thaliana* leaves, are of a hormetic type and that PSII of young leaves performs better than that of mature leaves under water deficit stress. The higher performance of PSII in young leaves was evident by the higher F*v*/F*m*, Φ*_PSII_*, ETR, and q*p*, at both mild water deficit stress (MiWDS) and moderate water deficit stress (MoWDS), compared to mature leaves. The U-shaped biphasic response curve of effective quantum yield of PSII photochemistry in *A. thaliana* young and mature leaves developed with a decrease at mild water deficit stress (MiWDS), which was followed by an increase in PSII photochemistry at moderate water deficit stress (MoWDS). The hormetic responses of PSII function are suggested to be triggered by the intensified ROS production in both young and mature leaves under MiWDS, which was considered to be beneficial for activating defense stress responses. We suggest that hormetic responses of PSII in *A. thaliana* under water deficit stress are regulated by the leaf developmental stage that modulates anthocyanin accumulation in a stress-dependent dose. Breeding of plants with high anthocyanin content that confers drought resilience could help crop production under future climate change.

## Figures and Tables

**Figure 1 ijms-24-09573-f001:**
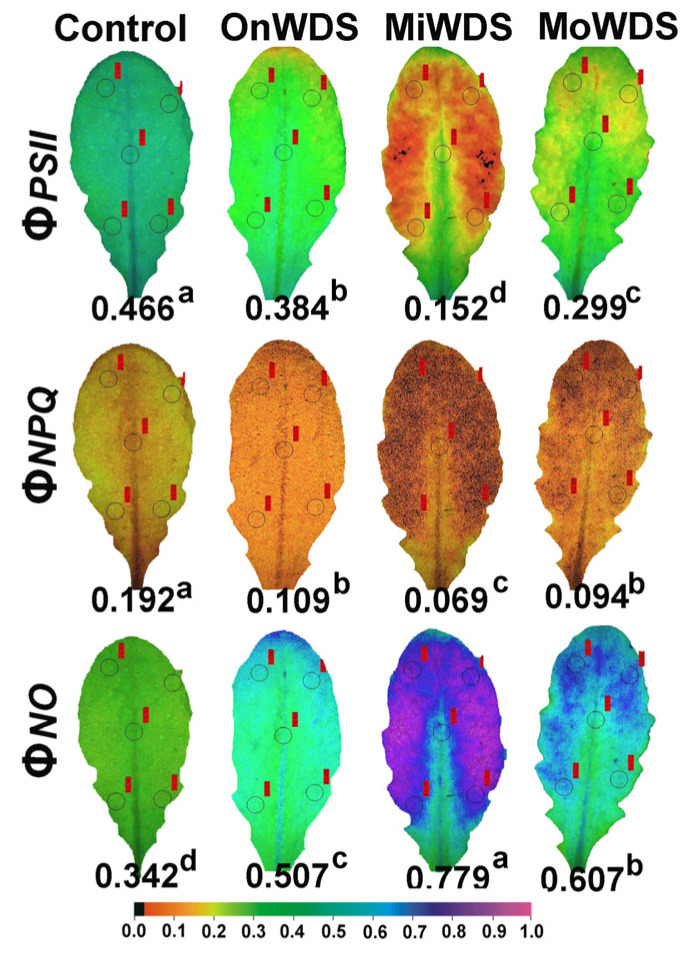
Representative color-coded leaf pictures of the light energy use in photosystem II. The effective quantum yield of PSII photochemistry (Φ*_PSII_*), the quantum yield of regulated non-photochemical energy loss in PSII (Φ*_NPQ_*), and the quantum yield of non-regulated energy loss in PSII (Φ*_NO_*) of *A. thaliana* mature leaves under optimal water conditions (control), at the onset of water deficit stress (OnWDS), at mild water deficit stress (MiWDS), and at moderate water deficit stress (MoWDS). Different lower-case letters for the same parameter indicate a significant difference between the water deficit stress treatments. A color code is shown at the bottom with values ranging from 0.0 to 1.0.

**Figure 2 ijms-24-09573-f002:**
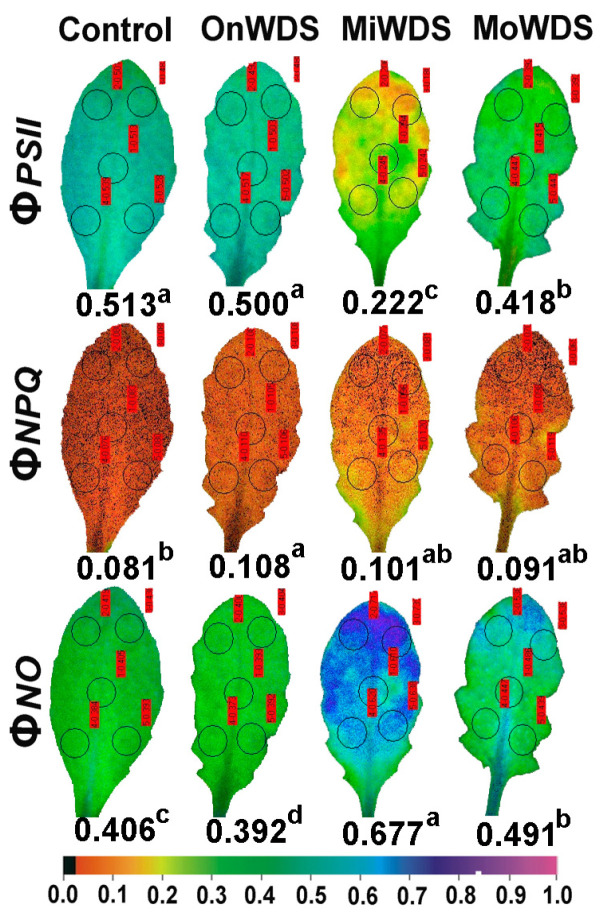
Representative color-coded leaf pictures of the light energy use in photosystem II. The effective quantum yield of PSII photochemistry (Φ*_PSII_*), the quantum yield of regulated non-photochemical energy loss in PSII (Φ*_NPQ_*), and the quantum yield of non-regulated energy loss in PSII (Φ*_NO_*) of *A. thaliana* young leaves under optimal water conditions (control), at the onset of water deficit stress (OnWDS), at mild water deficit stress (MiWDS), and at moderate water deficit stress (MoWDS). Different lower-case letters for the same parameter indicate a significant difference between the water deficit stress treatments. A color code is shown at the bottom with values ranging from 0.0 to 1.0.

**Figure 3 ijms-24-09573-f003:**
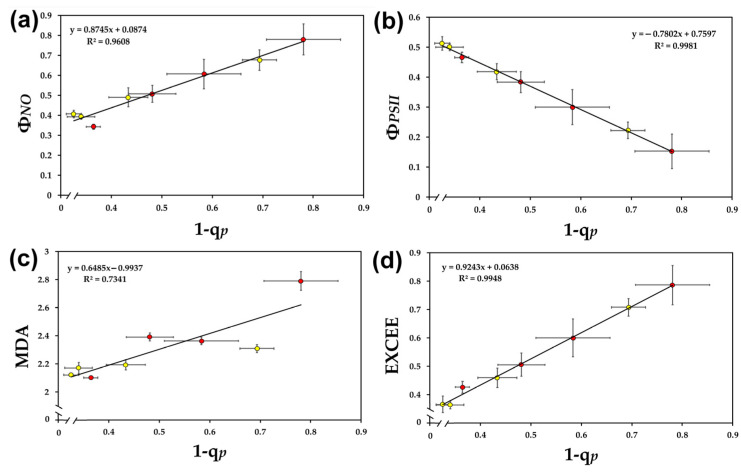
The relationships between the excitation pressure (1-q*_p_*) and the quantum yield of non-regulated energy dissipated in PSII (Φ*_NO_*) (**a**), the effective quantum yield of PSII photochemistry (Φ*_PSII_*) (**b**), the level of lipid peroxidation (MDA) (**c**), and the level of excess excitation energy (EXCEE) (**d**) in young (yellow) and mature (red) *A. thaliana* leaves at optimum water availability and under water deficit conditions. Error bars represent ± standard error of the mean (*n* = 3–5).

**Figure 4 ijms-24-09573-f004:**
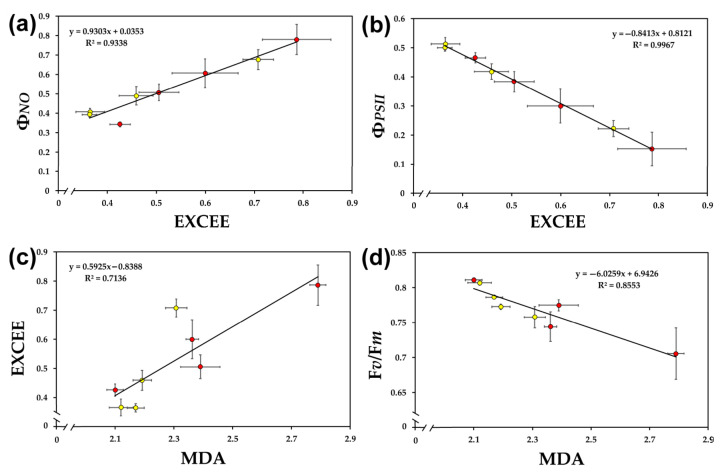
The relationships between the level of excess excitation energy (EXCEE) and the quantum yield of non-regulated energy dissipated in PSII (Φ*_NO_*) (**a**), the effective quantum yield of PSII photochemistry (Φ*_PSII_*) (**b**), the level of lipid peroxidation (MDA) (**c**), and the relationship between the level of lipid peroxidation (MDA) with the maximum efficiency of PSII photochemistry (F*v*/F*m*) (**d**) in young (yellow) and mature (red) *A. thaliana* leaves at optimum water availability and under water deficit conditions. Error bars represent ± standard error of the mean (*n* = 3–5).

**Figure 5 ijms-24-09573-f005:**
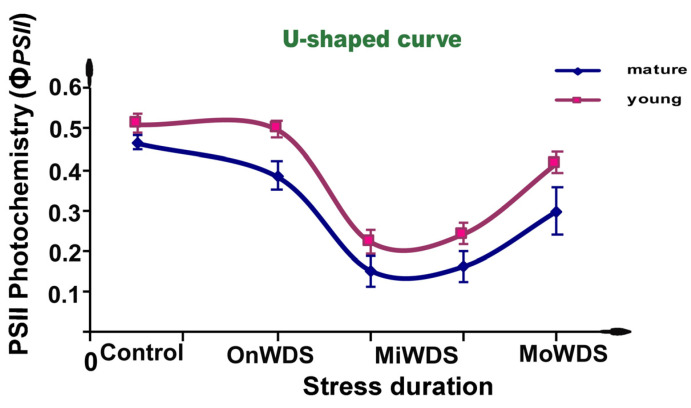
A U-shaped biphasic response curve of the effective quantum yield of PSII photochemistry (Φ*_PSII_*) of *A. thaliana* plants to water deficit stress. Control (i.e., optimal water availability), onset of water deficit stress (OnWDS), mild water deficit stress (MiWDS), prolonged MiWDS, and moderate water deficit stress (MoWDS).

**Table 1 ijms-24-09573-t001:** The leaf water content, the maximum efficiency of PSII photochemistry (F*v*/F*m*), the lipid peroxidation measured as malondialdehyde (MDA) content and representing oxidative stress, and the anthocyanin accumulation in young and mature *A. thaliana* leaves under optimal water conditions (control), at the onset of water deficit stress (OnWDS), and under mild water deficit stress (MiWDS) and moderate water deficit stress (MoWDS).

Parameter	Leaf Age	Control	OnWDS	MiWDS	MoWDS
Leaf Water Content (%)	Young	94.2 ± 0.2 ^a^	89.6 ± 0.2 ^b^	87.7 ± 0.4 ^c^ *	82.5 ± 0.2 ^d^ *
	Mature	94.7 ± 0.2 ^A^	89.9 ± 0.2 ^B^	84.2 ± 0.3 ^C^	79.4 ± 0.2 ^D^
F*v*/F*m*	Young	0.807 ± 0.003 ^a^	0.786 ± 0.002 ^b^	0.758 ± 0.015 ^d^ *	0.773 ± 0.004 ^c^ *
	Mature	0.811 ± 0.003 ^A^ *	0.775 ± 0.008 ^B^	0.705 ± 0.037 ^D^	0.744 ± 0.021 ^C^
MDA (nmol mL^−1^)	Young	2.12 ± 0.041 ^c^ *	2.17 ± 0.029 ^bc^	2.31 ± 0.036 ^a^	2.19 ± 0.031 ^b^
	Mature	2.10 ± 0.028 ^C^	2.39 ± 0.067 ^B^ *	2.79 ± 0.027 ^A^ *	2.36 ± 0.022 ^B^ *
Anthocyanins (Absorbance cm^−2^)	Young	0.097 ± 0.009 ^d^	0.105 ± 0.012 ^c^	1.961 ± 0.130 ^b^ *	3.247 ± 0.273 ^a^ *
	Mature	0.123 ± 0.012 ^D^ *	0.133 ± 0.016 ^C^ *	1.693 ± 0.115 ^B^	2.698 ± 0.263 ^A^

Different lower-case letters (a, b, c, d) in young leaves, and capital letters (A, B, C, D) in mature leaves, indicate a significant difference between the water deficit stress treatments in the same leaf developmental stage, while an asterisk (*) indicates significant difference between young and mature *A. thaliana* leaves, for the same water deficit stress treatment (*n* = 3–5).

**Table 2 ijms-24-09573-t002:** The electron transport rate (ETR), the redox state of the plastoquinone pool, representing the fraction of open PSII reaction centers (q*p*), the excess excitation energy at PSII (EXCEE), and the efficiency of the open PSII reaction centers (F*v*’/F*m*’) in young and mature *A. thaliana* leaves under optimal water conditions (control), at the onset of water deficit stress (OnWDS), at mild water deficit stress (MiWDS), and at moderate water deficit stress (MoWDS).

Parameter	Leaf Age	Control	OnWDS	MiWDS	MoWDS
ETR (μmol m^−2^ s^−1^)	Young	29.28 ± 1.28 ^a^ *	28.58 ± 0.64 ^a^ *	12.67 ± 1.57 ^c^ *	23.87 ± 1.50 ^b^ *
	Mature	26.59 ± 0.99 ^A^	21.92 ± 2.01 ^B^	8.71 ± 3.30 ^D^	17.08 ± 3.34 ^C^
q*p*	Young	0.674 ± 0.017 ^a^ *	0.660 ± 0.030 ^a^ *	0.306 ± 0.038 ^c^ *	0.566 ± 0.043 ^b^ *
	Mature	0.635 ± 0.014 ^A^	0.519 ± 0.047 ^B^	0.219 ± 0.073 ^D^	0.416 ± 0.074 ^C^
EXCEE	Young	0.365 ± 0.029 ^c^	0.364 ± 0.015 ^c^	0.708 ± 0.031 ^a^	0.459 ± 0.035 ^b^
	Mature	0.426 ± 0.021 ^D^ *	0.505 ± 0.041 ^C^ *	0.786 ± 0.069 ^A^ *	0.599 ± 0.067 ^B^ *
F*v*’/F*m*’	Young	0.760 ± 0.017 ^a^	0.759 ± 0.020 ^a^	0.724 ± 0.018 ^b^	0.739 ± 0.011 ^b^
	Mature	0.733 ± 0.013 ^A^	0.740 ± 0.003 ^A^	0.688 ± 0.033 ^B^	0.716 ± 0.014 ^AB^

Different lower-case letters (a, b, c) in young leaves, and capital letters (A, B, C, D) in mature leaves, indicate a significant difference between the water deficit stress treatments in the same leaf developmental stage, while an asterisk (*) indicates significant difference between young and mature *A. thaliana* leaves for the same water deficit stress treatment (*n* = 3–5).

## Data Availability

The data presented in this study are available in this article.

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
