# Peer review of "Hormesis Responses of Photosystem II in Arabidopsis thaliana under Water Deficit Stress"

_ijms, 2023, doi:10.3390/ijms24119573_

Round 1

Reviewer 1 Report

Dear Authors

Please find the attached file named; ijms-2401973-peer-review-v1111

Author Response

The responses are on the reviewers pdf file.

Reviewer 2 Report

This serves as my review report for the manuscript by Sperdouli et al. on the responses of Arabidopsis leaves to water deficit stress. The study is important to the field of plant stress biology. I however have several comments that need to be addressed by the authors.

1. The manuscript title needs to be shorted to improve clarity.

2. The entire manuscript requires extensive language editing to improve the quality of the writing, especially the abstract and introduction sections. Unfortunately, I am unable to list all areas that need correction.

3. I do not think that the study elucidated all mechanisms of drought tolerance in PSII. Therefore, please restructure the text in Lines 17-18.

4. The authors do not need to always write Arabidopsis thaliana in full in the manuscript. The authors can write the common name Arabidopsis and the scientific name Arabidopsis thaliana at first mention. After that, the common name can be used alone, unless it is necessary to use the scientific name. Also, A. thaliana can be used in subsequent text if required.

5. Please avoid writing lengthy sentences such as in Lines 24-28 and others, as clarity of meaning is reduced in such sentences.

6. The authors should note that lower levels of oxidative stress or oxidative damage in plant tissues are not only due to high levels of anthocyanins. The enzymatic and non-enzymatic antioxidant systems in plants are quite diverse and work together in regulating ROS  levels in tissues. Therefore, attributing the low lipid peroxidation levels to anthocyanins alone is only partially correct (Lines 22-28).

7. Furthermore, attributing the hormetic response of PSII to increased ROS under MiWDS is not entirely accurate (Lines 28-30). Stress response is a complex trait with many physiological and molecular processes contributing to stress adaptation. 

8. The introduction section has language errors that affect the scientific content and context of the writing. There are also several instances of repetition of text within and between paragraphs in some parts of the introduction e.g. in Lines 68-72; Lines 100-105; and so on. Please correct accordingly.

9. In Lines 62-67 please note the signalling functions of ROS are concentration-dependent, and this aspect needs to be mentioned somewhere in these sentences with appropriate reference citations.

10.  In line 114, "manipulated" should be replaced by "modulated".

11.  In Line 407, delete Pullman, WA, since these are stated in Line 406.

12.  If the young and mature leaves used in section 4.3 are similar to those described in Lines 396-5, do not repeat the leaf measurements in section 4.3.

13. I do not quite understand the use of small letters and CAPS to distinguish between the statistical analysis between the 2 leaf types. Is there no other way of presenting the stats analysis in a much clearer manner by using letters but not in small letters and CAPS ?

14. Is the MDA content between young and mature leafs significantly different  under control conditions  (Lines 140-142; Table 1)?

15. in lines 154-156, please revise the description of results as using the word "retained" implies no change in the measured parameter relative to the control.

16. move the last clause after the comma in Lines 156-157 to the discussion.

17. The results section is generally well-written. However, the authors should consistently remind the readers what the comparisons are. For example, in Lines 161-164, what are the measurements of water deficits being compared to?

18. The main legends in Figures 1 and 2 need revision to shorten them. Other descriptive texts can always be written after the main legend. 

19. Please correct the spelling of "reaction" in Line 197.

20. Please check if ETR is shown in Figure 1 as stated in Line 214.

21. Please double-check the text in Lines 217-218 relative to the measurements in Table 2. 

22.  Please double check  the font type/size used in the legend of Figure 3 is consistent.

23. Please double-check the figure parts cited in section 9. There could be an error in Line 250.

24. In Line 278, 287, please include the word possibly  after was.

25. In lines 270-297, please discuss your results relative to other published studies.

26. Unless there are functional studies included on the antioxidant nature of anthocyanins in this particular study, I would encourage the statement  in Line 307-310 to be written with caution. As stated earlier, the antioxidant machinery of plants is quite diverse, and attributing all reduction of ROS to anthocyanins is not entirely correct.

27. In Line 320, competence in what?

28. Figure 5 seems incomplete as there are no units of measurement on any of the axes. For example, where is the control or mild water deficit stress?

29. The U-shaped biphasic response curve mentioned in the conclusion should be well annotated for concluding remarks to be valid.

30. I am not sure I saw any mention of biological replicates in this study. Where there any biological replications at all used in this study?

The entire manuscript requires extensive language editing to improve the quality of the writing, especially the abstract and introduction sections. Unfortunately, I am unable to list all areas that need correction.

Author Response

This serves as my review report for the manuscript by Sperdouli et al. on the responses of Arabidopsis leaves to water deficit stress. The study is important to the field of plant stress biology. I however have several comments that need to be addressed by the authors.

  1. The manuscript title needs to be shorted to improve clarity.

The title was shortened.

  1. The entire manuscript requires extensive language editing to improve the quality of the writing, especially the abstract and introduction sections. Unfortunately, I am unable to list all areas that need correction.

Extensive language editing was conducted.

  1. I do not think that the study elucidated all mechanisms of drought tolerance in PSII. Therefore, please restructure the text in Lines 17-18.

The text in Lines 17-18 (now 16-17) was rewritten according to your comment.

  1. The authors do not need to always write Arabidopsis thaliana in full in the manuscript. The authors can write the common name Arabidopsis and the scientific name Arabidopsis thaliana at first mention. After that, the common name can be used alone, unless it is necessary to use the scientific name. Also, A. thaliana can be used in subsequent text if required.

Yes, we adopted this.

  1. Please avoid writing lengthy sentences such as in Lines 24-28 and others, as clarity of meaning is reduced in such sentences.

The lengthy sentence in Lines 23-27 was separated into two sentences. The same was followed in other lengthy sentences.

  1. The authors should note that lower levels of oxidative stress or oxidative damage in plant tissues are not only due to high levels of anthocyanins. The enzymatic and non-enzymatic antioxidant systems in plants are quite diverse and work together in regulating ROS  levels in tissues. Therefore, attributing the low lipid peroxidation levels to anthocyanins alone is only partially correct (Lines 22-28).

Yes, we agree that the levels of oxidative stress or oxidative damage in plant tissues are not only due to high levels of anthocyanins. In Lines 21-27 we do not attribute the level of oxidative stress to the level of anthocyanins.

  1. Furthermore, attributing the hormetic response of PSII to increased ROS under MiWDS is not entirely accurate (Lines 28-30). Stress response is a complex trait with many physiological and molecular processes contributing to stress adaptation. 

We agree with your comment that the stress response is a complex trait with many physiological and molecular processes contributing to stress adaptation. In Lines 28-30 we mention that “the hormetic response is suggested to be triggered” based on our results and the literature.

  1. The introduction section has language errors that affect the scientific content and context of the writing. There are also several instances of repetition of text within and between paragraphs in some parts of the introduction e.g. in Lines 68-72; Lines 100-105; and so on. Please correct accordingly.

Language errors in Introduction were corrected and the repetitions were deleted.

  1. In Lines 62-67 please note the signalling functions of ROS are concentration-dependent, and this aspect needs to be mentioned somewhere in these sentences with appropriate reference citations.

We discuss about the signaling functions of ROS in Discussion section  (Lines 494-503, 521-529) with the appropriate reference citations.

  1. In line 114, "manipulated" should be replaced by "modulated".

We replaced "manipulated" with "modulated".

  1. In Line 407, delete Pullman, WA, since these are stated in Line 406.

We deleted it.

  1. If the young and mature leaves used in section 4.3 are similar to those described in Lines 396-5, do not repeat the leaf measurements in section 4.3.

Yes, we deleted the leaf measurements in section 4.3.

  1. I do not quite understand the use of small letters and CAPS to distinguish between the statistical analysis between the 2 leaf types. Is there no other way of presenting the stats analysis in a much clearer manner by using letters but not in small letters and CAPS ?

We used lower-case (small) letters for the statistical analysis between the water deficit stress treatments inside the young leaves and capital letters (CAPS) for the statistical analysis between the water deficit stress treatments inside the mature leaves. An asterisk (*) indicates significant difference between young and mature leaves for the same water deficit stress treatment

  1. Is the MDA content between young and mature leafs significantly different  under control conditions  (Lines 140-142; Table 1)?

Yes, MDA content was significant higher in young leaves under control conditions as it is shown by the asterisk (*) in young leaves (2.12 ± 0.041c *)

  1. in lines 154-156, please revise the description of results as using the word "retained" implies no change in the measured parameter relative to the control.

We changed the word "retained" with "had ".

  1. move the last clause after the comma in Lines 156-157 to the discussion.

Yes, we moved it to Discussion (Lines 392-393).

  1. The results section is generally well-written. However, the authors should consistently remind the readers what the comparisons are. For example, in Lines 161-164, what are the measurements of water deficits being compared to?

In Lines 161-164 (now 225-238), the effective quantum yield of PSII photochemistry (ΦPSII) is compared between young and mature leaves, under all water deficit treatments.

  1. The main legends in Figures 1 and 2 need revision to shorten them. Other descriptive texts can always be written after the main legend.

We introduced a general main legend in Figures 1 and 2 and then we explained the main legend in details so that the Figures to be self-explanatory without the reader to have to consult information from the text.

  1. Please correct the spelling of "reaction" in Line 197.

Yes, we corrected it.

  1. Please check if ETR is shown in Figure 1 as stated in Line 214.

We corrected it. It was a mistake, it must be (Table 2).

  1. Please double-check the text in Lines 217-218 relative to the measurements in Table 2. 

Text in Lines 217-218 (now 304-305) was rewritten.

  1. Please double check  the font type/size used in the legend of Figure 3 is consistent.

Thank you for pointing it. It was corrected

  1. Please double-check the figure parts cited in section 9. There could be an error in Line 250.

We corrected it. It was a mistake between 4c and 4d.

  1. In Line 278, 287, please include the word possibly after was.

We inserted the word possibly in both cases (now 385,405).

  1. In lines 270-297, please discuss your results relative to other published studies.

Results in lines 270-279 (now 374-417) were discussed relatively to other studies.

  1. Unless there are functional studies included on the antioxidant nature of anthocyanins in this particular study, I would encourage the statement  in Line 307-310 to be written with caution. As stated earlier, the antioxidant machinery of plants is quite diverse, and attributing all reduction of ROS to anthocyanins is not entirely correct.

We have rewritten this paragraph (now Lines 432-438)

  1. In Line 320, competence in what?

We have rewritten the sentence (now Lines 451-454).

  1. Figure 5 seems incomplete as there are no units of measurement on any of the axes. For example, where is the control or mild water deficit stress?

We have revised Figure 5 and transferred it to results section (2.10).

  1. The U-shaped biphasic response curve mentioned in the conclusion should be well annotated for concluding remarks to be valid.

We extended the annotation of the U-shaped biphasic response curve describing Figure 5 in the results section (2.10).

  1. I am not sure I saw any mention of biological replicates in this study. Where there any biological replications at all used in this study?

Biological replicates were mentioned in Figures 3 and 4 and now have been included in Tables 1 and 2.

Comments on the Quality of English Language

The entire manuscript requires extensive language editing to improve the quality of the writing, especially the abstract and introduction sections. Unfortunately, I am unable to list all areas that need correction.

Extensive language editing was conducted.

Reviewer 3 Report

Dear authors,

Your manuscript is worked on, but to increase its value, I recommend some improvements that can be found directly on the attached manuscript.

Lines 113-117: The last sentence is too long. You can separate the following sentence from the rest: <Thus, in order to address this issue, we explored how anthocyanin accumulation and lipid 113 peroxidation are manipulated in young and mature Arabidopsis thaliana leaves, at the onset 114 of water deficit stress (OnWDS), mild water deficit stress (MiWDS), and moderate water 115 deficit stress (MoWDS)>

Line 113: Related to: <...we explored how anthocyanin accumulation... > it would be good to say what you intend to do using the verb in the future tense, like: <...we set out to explore... >. All this to create interest for readers. As it is, it seems to be something ready made as a conclusion.

Line 117: I suggest you briefly explain why you chose the plant Arabidopsis thaliana for study?

Tabe 1: a. I suggest that you mention what a,b,c,d represent where it suits you? You should also mention where they come from. If I'm talking about statistics, it would be good to tell about the method.

b. Is it advisable to explain what Fv/Fm means? Explain in the footer of the table and all the abbreviations not covered.

Lines 120-124: I recommend expanding the textual explanations of table 1, as it is not clear what data you are actually referring to.

Lines 398-396: Some aspects need to be clarified in Material and method. For example: The first time you talk about <... seedlings of A. thaliana.. >, then you say that ... < For uniform germination, the seeds were incu-393 bated for 2 days at 4 °C before sowing on soil>.

Lines 390-393. It is not clear how you proceeded, seeded or planted? Did you use seeds or seedlings?

Where did you get the material from?

How many plants did you use in the study?

What variety did you use and how does it lend itself to drought conditions? Characterize in short words the variety/hybrid used.

Line 460: I suggest you add and justify the study by giving it a practical applicability. I mean, why is it important and for which categories of plants is it relevant? In which fields can it be applied, I suspect that especially in agriculture, biology and others. Whatever the explanation, you must give value to the work.

 R

Author Response

Your manuscript is worked on, but to increase its value, I recommend some improvements that can be found directly on the attached manuscript.

Lines 113-117: The last sentence is too long. You can separate the following sentence from the rest: <Thus, in order to address this issue, we explored how anthocyanin accumulation and lipid 113 peroxidation are manipulated in young and mature Arabidopsis thaliana leaves, at the onset 114 of water deficit stress (OnWDS), mild water deficit stress (MiWDS), and moderate water 115 deficit stress (MoWDS)>

We split the last long sentence into two as you suggested.

Line 113: Related to: <...we explored how anthocyanin accumulation... > it would be good to say what you intend to do using the verb in the future tense, like: <...we set out to explore... >. All this to create interest for readers. As it is, it seems to be something ready made as a conclusion.

The sentence was rewritten as you suggested.

Line 117: I suggest you briefly explain why you chose the plant Arabidopsis thaliana for study?

We explain (lines 154-156, 542-544) that we have chosen the plant Arabidopsis thaliana for our study since it is considered as model plant.

Tabe 1: a. I suggest that you mention what a,b,c,d represent where it suits you? You should also mention where they come from. If I'm talking about statistics, it would be good to tell about the method.

We explain at the footer of the tables what a,b,c,d represent, while the method of statistics is described in section 4.6.

Is it advisable to explain what Fv/Fm means? Explain in the footer of the table and all the abbreviations not covered.

Fv/Fm and the other abbreviations are explained in the Table caption.

Lines 120-124: I recommend expanding the textual explanations of table 1, as it is not clear what data you are actually referring to.

In lines 120-124 (now 163-167) we are referring to the leaf water content (%)

Lines 398-396: Some aspects need to be clarified in Material and method. For example: The first time you talk about seedlings of A. thaliana, then you say that For uniform germination, the seeds were incubated for 2 days at 4 °C before sowing on soil.

Lines 390-393. It is not clear how you proceeded, seeded or planted? Did you use seeds or seedlings?

We have rewritten the text as follows (lines 541-548) “A. thaliana (L.) Heynh. (Col-0) seeds obtained from Nottingham Arabidopsis Stock Centre (NASC) were sown on a soil and peat mixture in a growth chamber [114]. For uniform germination, the seeds were incubated for 2 days at 4 °C before sowing on soil. The germinated seedlings of A. thaliana ecotype Columbia (Col-0) were grown in a growth chamber with a long day photoperiod 14 h/10 h, temperature 23 ± 1/20 ± 1 °C day/night, humidity 45 ± 5/60 ± 5 % day/night, and light intensity of 130 ± 10 μmol photons m−2 s−1.

Where did you get the material from?

We get the seeds from Nottingham Arabidopsis Stock Centre (NASC)

How many plants did you use in the study?

We used 40 plants in the study. Biological replicates were mentioned in Figures 3 and 4, and now have been included in Tables 1 and 2.

What variety did you use and how does it lend itself to drought conditions? Characterize in short words the variety/hybrid used.

We mention in materials and methods that we used Arabidopsis thaliana ecotype Columbia (Col-0) as plant material.

Line 460: I suggest you add and justify the study by giving it a practical applicability. I mean, why is it important and for which categories of plants is it relevant? In which fields can it be applied, I suspect that especially in agriculture, biology and others. Whatever the explanation, you must give value to the work.

A sentence with a practical application in agriculture under the future climate change was added (line 653-654).

Round 2

Reviewer 2 Report

I thank the authors for their revision. A few comments below:

1. Please recheck the structure and intended meaning of lines 23-25.

2. in line 45, the word "outcomes" needs revision.

3. In Lines 63-66, please consider rewording as " Moreover, ............photosynthesis, resulting in stomatal closure..........., thus limiting CO2.........

Author Response

Thank you for your comments that helped us to improve our manuscript. All new changes were highlighted in yellow.

  1. Please recheck the structure and intended meaning of lines 23-25.

The sentence was rewritten.

  1. in line 45, the word "outcomes" needs revision.

The word "outcomes" was replaced with "results".

  1. In Lines 63-66, please consider rewording as " Moreover, ............photosynthesis, resulting in stomatal closure..........., thus limiting CO2.........

We rephrased the sentence as you suggested.